# Molecular Mechanism of Food-Derived Polyphenols on PD-L1 Dimerization: A Molecular Dynamics Simulation Study

**DOI:** 10.3390/ijms222010924

**Published:** 2021-10-10

**Authors:** Yan Guo, Jianhuai Liang, Boping Liu, Yulong Jin

**Affiliations:** Key Laboratory for Bio-Based Materials and Energy of Ministry of Education, College of Materials and Energy, South China Agricultural University, Guangzhou 510630, China; guoyan@stu.scau.edu.cn (Y.G.); j.h_liang@stu.scau.edu.cn (J.L.)

**Keywords:** PD-1/PD-L1 pathway, food-derived polyphenols, small-molecule inhibitors, molecular docking, molecular dynamics simulation

## Abstract

In cancer immunotherapy, an emerging approach is to block the interactions of programmed cell death-1 (PD-1) and programmed cell death-ligand 1 (PD-L1) using small-molecule inhibitors. The food-derived polyphenols curcumin (CC), resveratrol (RSV) and epigallocatechin gallate (EGCG) have anticancer immunologic functions, which, recently, have been proposed to act via the downregulation of PD-L1 expression. However, it remains unclear whether they can directly target PD-L1 dimerization and, thus, interrupt the PD-1/PD-L1 pathway. To elucidate the molecular mechanism of such compounds on PD-L1 dimerization, molecular docking and nanosecond molecular dynamics simulations were performed. Binding free energy calculations show that the affinities of CC, RSV and EGCG to the PD-L1 dimer follow a trend of CC > RSV > EGCG. Hence, CC is the most effective inhibitor of the PD-1/PD-L1 pathway. Analysis on contact numbers, nonbonded interactions and residue energy decomposition indicate that such compounds mainly interact with the C-, F- and G-sheet fragments of the PD-L1 dimer, which are involved in interactions with PD-1. More importantly, nonpolar interactions between these compounds and the key residues Ile54, Tyr56, Met115, Ala121 and Tyr123 play a dominant role in binding. Free energy landscape and secondary structure analyses further demonstrate that such compounds can stably interact with the binding domain of the PD-L1 dimer. The results provide evidence that CC, RSV and EGCG can inhibit PD-1/PD-L1 interactions by directly targeting PD-L1 dimerization. This provides a novel approach to discovering food-derived small-molecule inhibitors of the PD-1/PD-L1 pathway with potential applications in cancer immunotherapy.

## 1. Introduction

Programmed cell death-1 (PD-1) [1,2,3] is an immune checkpoint protein that can be expressed on the surface of activated immune cells. Its corresponding ligand, programmed cell death ligand-1 (PD-L1), is overexpressed in many kinds of cancer. When PD-1 binds to PD-L1, immune cells are inhibited from attacking cancer cells; hence, blocking the interaction between PD-1 and PD-L1 is a promising approach to cancer immunotherapy [4]. The pioneering work in this field has mainly focused on monoclonal antibodies (mAbs). Several mAbs, such as nivolumab, avelumab and atezolizumab, have been proven to interrupt the PD-1/PD-L1 pathway and are at the stage of clinic application or approval [5,6,7,8,9]. Recently, greater attention has been paid to small-molecule inhibitors (MW < 550 Da) as they have higher stability and better tumor penetration than macromolecular mAbs [10]. Remarkable progress has been made by the Bristol Myers Squibb (BMS) company, which has synthesized a series of small molecules that can disrupt PD-1/PD-L1 interactions by inducing PD-L1 dimerization and binding to the inner surface of the PD-L1 dimer [11,12,13].

In addition to these synthetic small-molecule inhibitors, numerous naturally available food-derived compounds such as polyphenols have also received much attention owing to their lower toxicity and fewer side effects. Indeed, some have already been experimentally verified to be active in blocking the PD-1/PD-L1 pathway [14,15,16]. Rawangkan et al. [17] found that epigallocatechin gallate (EGCG), which is the main constituent of green tea catechins, can inhibit PD-L1 expression in non-small-cell lung cancer cells. Another polyphenol, curcumin (CC), which is derived from the root of *Curcuma longa* L. [18,19], has also been widely investigated and proven to be a cancer immunomodulatory molecule due to its inhibitory effect on PD-L1 expression [15,20,21,22,23,24,25]. Chen et al. [26] and Chin et al. [27] evaluated the effects of resveratrol (RSV), which is commonly present in red grape skin and berries, on oral and ovarian cancer cells, respectively. It was shown to be efficacious in the downregulation of PD-L1 expression. Very recently, Verdura et al. [28] found that RSV is able to directly target PD-L1 dimerization to enhance anti-tumor T-cell immunity. This is an inhibitory mechanism known to occur with BMS’s synthetic small molecules, but it has rarely been reported in natural substances [11,12,13]. This suggests that CC and EGCG might also be able to block the PD-1/PD-L1 pathway by directly binding to the PD-L1 dimer.

The current study is a systematic investigation of the molecular mechanisms of RSV, CC and EGCG on the PD-L1 dimer, which was performed via molecular modeling. Based on the disclosed X-ray crystal structure of the PD-L1 dimer in complex with the inhibitor BMS-200 (PDB ID: 5N2F) and the three-dimensional (3D) structures of RSV (PDB ID: 1CGZ), CC (PDB ID: 4K58) and EGCG (PDB ID: 3NG5), the docking of such food-derived compounds to the PD-L1 dimer was first conducted to obtain complex systems (see Figure 1 and Figure 2). Then, a combination of molecular dynamics (MD) simulation and molecular mechanics Poisson–Boltzmann surface area (MM-PBSA) approaches [29,30,31,32,33] was used to explore the energy contribution of the interfacial residues on the PD-L1 dimer. To explore the binding modes between the PD-L1 dimer and such food-derived compounds, contact numbers and nonbonded interactions were also determined. Further analysis of the free energy landscape (FEL) and secondary structure was applied to reflect the effects of such food-derived compounds on the overall dynamic characteristics of the PD-L1 dimer. Generally, our results confirm that the ability of such food-derived compounds to interrupt the PD-1/PD-L1 pathway by targeting PD-L1 dimerization is not unique to RSV, but is also found in CC and EGCG, although different efficacies were observed in terms of binding free energies. Furthermore, the binding regions of such compounds on the PD-L1 dimer are clarified, and the key residues and interactions involved in binding are highlighted. This work offers a new perspective on the potential to use such polyphenols in cancer immunotherapy and provides a clue toward the discovery of food-derived small-molecule inhibitors that have high efficiency.

## 2. Results and Discussion

It has been reported that RSV, CC and EGCG can inhibit PD-L1 expression, thus performing an important role in cancer immunotherapy. In particular, RSV is predicted to induce PD-L1 dimerization and interact with the inner surface of the PD-L1 dimer. Whether the compounds CC and EGCG can also directly bind to PD-L1 in a similar way as RSV, thus inhibiting PD-1/PD-L1 interactions, remains to be illuminated. In order to clarify this issue, a series of molecular modeling approaches were used in the present work, including molecular docking, MD simulations and MM-PBSA calculations.

### 2.1. Docking

The 3D structures of food-derived compounds and initial complex structures for MD simulations are displayed in Figure 1 and Figure 2. Molecular docking was conducted to provide initial models of the complex systems for subsequent MD simulation studies [34]. The X-ray crystal structure of the PD-L1 dimer (PDB ID 5N2F) was used as the receptor for this study. Validation of the docking protocol was performed by redocking the ligand BMS-200 (5N2F) into the PD-L1 dimer. As presented in Appendix A, both the crystal and docked structures overlapped within the cavity formed between _A_PD-L1 and _B_PD-L1, and the overlap showed a root mean square deviation (RMSD) of 0.77 Å. In addition to pose selection, interaction determination is another method of validation. The results show that the docked BMS-200 formed hydrophobic interactions with the residues _A_Met115, _A_Ala121, _A_Tyr123, _B_Ile54, _B_Tyr56, _B_Ala121 and _B_Tyr123 of the PD-L1 dimer. Moreover, it formed hydrogen bonds (H bonds) with _B_Gln66 and _A_Asp122 as well as a Π-stacking interaction with _B_Tyr56 (Appendix A). This suggested that the docking protocol could be utilized to identify the binding conformations of the CC, RSV and EGCG systems. Then, the docked complexes were used as initial coordinates in the MD simulations.

### 2.2. RMSD

Calculation of the RMSD over the residues within 20 Å of the ligand was performed to explore the structural stability of the systems over a duration of 150 ns [35]. As depicted in Figure 3a, there was not much deviation in the complex systems throughout the simulation time. This also shows that all complex systems were equilibrated after 5 ns and possessed relatively constant values of 2.30, 2.44 and 2.90 Å for the CC, RSV and EGCG systems, respectively. However, the PD-L1 dimer system had considerable deviations and attained stability at around 3.11 Å until 60 ns. This indicates that the complex systems possessed higher structural stability than the PD-L1 dimer system, which is attributed to the binding of such food-derived compounds. Moreover, the EGCG system was observed to have more flexible behaviors than those of the CC and RSV systems and, thus, the lowest capability to stabilize the PD-L1 dimer. Briefly, all the systems reached stable levels and could be utilized for further analysis.

### 2.3. RMSF

RMSF analysis was performed to reveal the per-residue fluctuations of the systems during the MD simulations. As illustrated in Figure 3b,c, all the systems followed a more or less similar trend in RMSF with minute exceptions in a few cases. Unsurprisingly, the most stable regions mainly occurred at the β-sheets (namely sheet A–G), in which the RMSF values only reached ~2 Å, and loops connected these regions and the helix domain. The most flexible segments were the *N*-terminal, *C*-terminal and loop regions. The BC loop of the dimer system showed an obviously higher RMSF value, reaching ~5 Å. In addition, it can also be found that the complex systems obtained higher residue stability relative to the dimer system. Analysis of RMSF of the three complex systems showed that the BC loop of the EGCG system kept a higher RMSF value (~4 Å), which is in accordance with the previous RMSD results. Moreover, comparative analysis of RMSFs among the β-sheet segments of the complex systems showed that residues 54–59, residues 110–117 and residues 121–124 (existing in the C, F and G segments, respectively) possessed a relatively low value of 1 Å, indicating that these regions were more stable due to the binding of the food-derived compounds.

### 2.4. Binding Free Energy

The above analysis demonstrates that the PD-L1 dimer would undergo conformational change when binding with such food-derived compounds. However, this work is focused much more on the ranking of binding affinities of the three complex systems. Hence, the binding free energies were calculated using the MM-PBSA approach based on 300 snapshots sampled from the last 30 ns of stable MD trajectories of each system. As depicted in Table 1, the average values of the binding free energies for the PD-L1 dimer with CC, RSV and EGCG were −33.72, −28.49 and −20.31 kcal/mol, respectively, suggesting that compound CC possesses the strongest binding affinity with the PD-L1 dimer. However, the mean value of Δ*G* for the PD-L1 dimer system was positive (36.11 kcal/mol), indicating that PD-L1 can hardly be spontaneously dimerized at all, and that the food-derived molecules are essential for PD-L1 dimerization. Moreover, the MM-PBSA method can also decompose the total binding free energy into individual components, thereby helping us to understand which interaction energies are significant in the binding process. For all three complex systems, the total nonpolar binding free energy represents the main driving force of binding of the food-derived compounds with the PD-L1 dimer. This was particularly manifested in the CC system. The Δ*E*_polar,total_ impedes binding, which is partially compensated for by the favorable electrostatic interaction energy Δ*E*_ele_. This was most evident in the EGCG system, which had an Δ*E*_ele_ value (−23.14 kcal/mol) approximately 18 kcal/mol stronger than those of the CC and RSV systems, probably due to its larger amount of hydroxyl groups (Figure 1).

### 2.5. Per-Residue Energy Decomposition

Next, the energy decomposition was performed to identify the residues making significant contributions to binding with the food-derived compounds. Here, the residues with energy contributions <−1 kcal/mol were considered to be key residues. As depicted in Figure 4, CC interacted with eight key residues: _A_Ile54, _A_Tyr56, _A_Met115, _A_Tyr123, _B_Ile54, _B_Tyr56, _B_Met115 and _B_Ala121. RSV also interacted with eight key residues: _A_Ile54, _A_Tyr56, _A_Met115, _A_Ile116, _B_Ile54, _B_Tyr56, _B_Met115 and _B_Ala121. Meanwhile, EGCG interacted with only five key residues: _A_Tyr123, _B_Ile54, _B_Tyr56, _B_Val68 and _B_Met115. Briefly, key residues Ile54, Tyr56, Met115, Ala121 and Tyr123 on either _A_PD-L1 or _B_PD-L1 participated in interactions with food-derived molecules. These almost entirely occupy the target space of BMS’s small molecules that bind to the PD-L1 dimer, such as BMS-200, BMS-8 and BMS-1166 [11,12,13]. More importantly, these residues are involved in the formation of the PD-1/PD-L1 interface. The energy decomposition results also demonstrate that Met115 makes obvious and constant contributions to ligand binding, which is consistent with the literature [36,37,38], indicating that the presence of small-molecule inhibitors could promote the conformational change of Met115, giving it improved accessibility to PD-L1 dimer binding sites. In addition, the key residues of _A_PD-L1 contributed approximately the same percentage of binding free energy in the CC and RSV systems as did _B_PD-L1, while EGCG was more inclined to bind with _B_PD-L1.

### 2.6. Contact Numbers

To characterize the binding regions of CC, RSV and EGCG on the PD-L1 dimer, the average contact numbers between the food-derived molecules and individual residues were computed (Figure 5). Herein, 10 contacts were considered as criteria to identify residues with significant effects on intermolecular interactions [39]. From Figure 5, it can be seen that CC preferred to interact with Ile54, Val55, Tyr56, Gln66, Val68, Met115, Ile116, Ser117, Ala121, Asp122 and Tyr123 on both PD-L1 monomers; RSV exhibited strong preferential interactions with residues Ile54, Val55, Tyr56, Met115, Ile116, Ser117, Ala121, Asp122 and Tyr123; while EGCG preferentially bound to distinct sites of _A_PD-L1 and _B_PD-L1 consisting of _A_Phe19, _A_Thr20, _A_Met115, _A_Ala121, _A_Asp122, _A_Tyr123, _A_Lys124, _B_Ile54, _B_Val55, _B_Tyr56, _B_Gln66, _B_Val68, _B_Met115, _B_Ile116 and _B_Ser117. In brief, these compounds interacted strongly with the C sheet (residues 54–56), F sheet (residues 115–117) and G sheet (residues 121–124) of the PD-L1 dimer. Moreover, both CC and EGCG bound to an additional region C’ sheet (residues 66–68) and EGCG bound to the *N*-terminal domain (residues 18–20). These results match well with the calculated free energies as presented in Table 1 and Figure 4.

The corresponding contact numbers of these regions are listed in Table 2. It can be seen that the total contact numbers of the CC system were significantly larger than those of the RSV system, while the EGCG system obtained an intermediate value, which may be due to the higher contact numbers in the *N*-terminal, C’ sheet and G sheet regions. In particular, the β-sheet fragments contributed to the majority of contact numbers of the complex systems, although those of the RSV system were slightly fewer. As illustrated in Figure 6, the most appealing feature is the strong hydrophobic interactions between this domain of the PD-L1 dimer and the benzene rings of CC, RSV and EGCG. In addition, it should be noted that the residues _B_Gln66, _B_Ser117 and _B_Asp122 can interact with the O atoms of CC through H bonds. The Π-stacking interaction between the sidechain of _B_Tyr56 and the benzene ring of CC further stabilizes the CC system. Furthermore, the residues _A_Tyr56, _A_Met115 and _B_Tyr123 form H bonds with the hydroxyl groups of RSV. Hence, greater attention should be paid to the multiple H bonds between EGCG and residues _A_Phe19, _A_Thr20, _A_Ala121, _A_Asp122, _A_Lys124, _B_Gln66 and _B_Met115. This is consistent with the binding free energy results showing that the electrostatic energy (−23.14 kcal/mol) of the EGCG system was significantly stronger relative to those of the other two complex systems. Regarding the *N*-terminal domain, _A_Phe19 and _A_Thr20 bind to EGCG by H bonds; however, such H-bond interactions could not be detected in this domain of the CC and RSV systems.

In short, the food-derived compounds with high inhibitory activities were able to form multiple hydrophobic interactions and H bonds with the PD-L1 dimer in the binding pockets.

### 2.7. Nonbonded Interactions

To further scrutinize the interactions anchoring the food-derived compounds to the binding sites, a quantitative evaluation of the occupancy of intermolecular H bonds along the entire MD simulation was performed (Table 3). The three complex systems showed obviously different patterns of H bonds. The O1 atom of CC forms an H bond (occupancy = 76.74%) with _B_Ser117; further, an H bond with an occupancy of 87.04% could also be observed between _B_Gln66 and the O2 atom of CC. In addition, _A_Met115 interacted with the O3 atom of RSV by an H bond, and the occupancy value (76.74%) further demonstrates its stability throughout the simulation. Nevertheless, EGCG formed multiple H bonds with the O atom of _A_Ala121, _A_Phe19 and _B_Met115, as well as the OD1 and N atoms of the negatively charged residue _A_Asp122. In short, the EGCG system displayed more H bonds than the CC and RSV systems but had the lowest binding free energy value, suggesting that H bonds perhaps do not play a dominant role in ligand binding, although several polar residues, such as Thr20, Gln66, Asp122 and Lys124, could be observed in the pocket.

Next, the detailed binding modes of the complex systems were investigated, and the relevant results are illustrated in Figure 6. As shown, the food-derived compounds were located in a linear cylindrical tunnel between _A_PD-L1 and _B_PD-L1. The pocket of the CC system was observed to be surrounded by the sidechains of residues _A_Ile54, _A_Tyr56, _A_Met115, _A_Ala121, _A_Tyr123, _B_Ile54, _B_Tyr56, _B_Gln66, _B_Met115, _B_Ser117, _B_Ala121 and _B_Asp122. The sidechains of _A_Tyr123, _B_Tyr54, _B_Met115 and _B_Ser117 extended toward the atoms C13, C2, C11 and O1 of CC, respectively; other residues approached the benzene rings of CC. The binding pocket of the RSV system was surrounded by residues _A_Ile54, _A_Tyr56, _A_Met115, _A_Ile116, _A_Ala121, _B_Ile54, _B_Tyr56, _B_Met115, _B_Ala121 and _B_Tyr123. In detail, the sidechain of _A_Ala121 was close to the C7 atom of RSV, while the sidechains of other residues were adjacent to the benzene rings of RSV. The pocket of the EGCG system was surrounded by residues _A_Phe19, _A_Thr20, _A_Ala121, _A_Asp122, _A_Tyr123, _A_Lys124, _B_Ile54, _B_Tyr56, _B_Gln66, _B_Val68 and _B_Met115, among which the sidechain of _B_Gln66 was near the O37 atom of EGCG and the others neighbored the aromatic rings of EGCG. Briefly, the interaction modes between the PD-L1 dimer and food-derived compounds were in good concordance with those of the PD-L1 dimer and BMS small-molecules, as well as the [1,2,4] triazolo [4,3-a] pyridines inhibitors [40,41,42,43,44,45]. In particular, the residues located at the binding pockets—Ile54, Tyr56, Met115, Ala121 and Tyr12—were highly conservative. Along with the aforementioned energy decomposition analysis, the results showed that these residues make significant contributions to binding with such compounds (Figure 4), which signifies that they are potential targets for the development of efficient food-derived drugs that can block the PD-1/PD-L1 pathway. In addition, it was also found that, compared to H bonds, the ligand binding was more reliant on hydrophobic interactions between the aromatic rings of the food-derived compounds and the hydrophobic residues _A_Met115, _A_Ala121, _B_Ile54, _B_Val68, _B_Met115 and _B_Ala121. Hence, structural modification of these aromatic rings would be a good approach to improving the efficiency of inhibiting PD-1/PD-L1 interactions by using food-derived compounds.

### 2.8. Cross-Correlation Matrix Analysis

To monitor the effect of the food-derived compounds on the correlated motion of the PD-L1 dimer more accurately, correlation matrix analysis was carried out [46]. In Figure 7, the depth of color on the diagonal indicates the degree of motion of the residues [32]. The positive regions in the matrices (red) represent the correlated movements of residues, while the negative regions (blue) signify the strong anticorrelations with residue motion. As illustrated, the residues on the diagonals, especially those on the *C*-terminal (i.e., residues 132–143 and 241–250 corresponding to residues 132–143 of _A_PD-L1 and residues 132–141 of _B_PD-L1, respectively), have stronger motions, which could be attributed to the high flexibility of the residues on this domain. It should also be noted that the blue patches fill a large part of the matrix in the dimer system, indicating anticorrelated motions of the amino acid residues between _A_PD-L1 and _B_PD-L1. However, a loss of those anticorrelated movements can be observed in some regions of the CC system, which might be related to the binding of CC. The anticorrelated movements in the RSV system are stronger than those in the CC system, which may help to explain why the activity of RSV is lower than that of CC. EGCG can also weaken the anticorrelated movement of the PD-L1 dimer, but with lower capacity than both RSV and CC. In short, this suggests that the complex systems exhibited more stable dynamic behaviors than the dimer system, owing to the linkages of the food-derived compounds. The cross-correlation matrix results are coincident with the binding free energy calculations, both of which indicate that the affinity of these compounds against the PD-L1 dimer follows a trend of CC > RSV > EGCG.

### 2.9. Free Energy Landscape

To further investigate the effect of the food-derived compounds on the conformational space of the PD-L1 dimer, their FELs were comparatively analyzed (Figure 8). An FEL was constructed by projecting the trajectories on the first two principal components (dihedral PC1 and PC2), which are considered as reaction coordinates. From the FELs, one major region was identified for the PD-L1 dimer in the presence of CC, while more major conformational regions were recognized for the PD-L1 dimer system, indicating that CC can stabilize the conformation of the PD-L1 dimer. However, upon the introduction of RSV, two major conformational regions were observed. In comparison, the conformational space of the PD-L1 dimer bound with EGCG involved four regions, suggesting that EGCG can induce three more conformational regions than CC and two more conformational regions than RSV. This indicates that EGCG has the lowest ability to stabilize the PD-L1 dimer and, thus, the lowest inhibitory activity. To sum up, the FELs reveal that the ability of such compounds to stabilize the PD-L1 dimer followed a CC > RSV > EGCG trend, which was also in good agreement with the calculated binding free energies. Hence, among the three food-derived compounds, CC was found to be the most effective inhibitor of the PD-1/PD-L1 pathway and, hence, has the greatest potential for use in cancer immunotherapy.

### 2.10. Secondary Structure

Finally, the secondary structures of the CC, RSV, EGCG and PD-L1 dimer systems were analyzed by utilizing the dictionary of secondary structure for proteins (DSSP) program implemented in GROMACS. As depicted in Figure 9, the secondary structures of the dimer system were mostly composed of β-sheet conformations, such as residues 37–42, 93–100 and 104–114 (corresponding to residues 54–59, 110–117 and 121–123 of _A_PD-L1). Residues 32–35 (corresponding to residues 49–52 of _A_PD-L1) exhibited alternating 3-helix and turn conformations throughout the simulation, with the 3-helix conformation being observed more frequently, while residues 158–161 (corresponding to residues 49–52 of _B_PD-L1) exhibited the opposite pattern. Residues 72–77 and 198–204 (corresponding to residues 89–94 of _A_PD-L1 and 89–95 of _B_PD-L1, respectively) possessed conformational changes of 3-helix, α-helix and turn throughout the simulation periods. Residues 116–126 (corresponding to residues 133–143 of _A_PD-L1) exhibited the α-helix conformation, which was transformed into turn, 3-helix, and coil conformations after 84 ns of simulation time. Similarly, residues 242–249 (corresponding to residues 133–140 of _B_PD-L1) also showed these conformations throughout the simulation, which were lost at one point and regained at some other points, indicating that the C-terminal was not stable.

The secondary structures of the complex systems (Figure 10, Figure 11 and Figure 12) showed similar phenomena to the PD-L1 dimer system, where the β-sheet domain was maintained throughout the MD simulation even when local dissimilarities existed as described above. Notably, the C, F and G sheet regions are essential to drug discovery, since they are crucial regions for interaction with PD-1. Thus, these compounds bind stably to these regions of the PD-L1 dimer, thereby interrupting PD-1/PD-L1 interactions. This implies that such compounds could be potential small-molecule drugs for targeting PD-L1.

## 3. Materials and Methods

### 3.1. Molecular Docking

Molecular docking is a tool for predicting ligands’ binding affinities to target proteins and exploring their possible binding modes [34]. The crystal structures of the PD-L1 dimer (PDB ID: 5N2F), CC (PDB ID: 4K58), RSV (PDB ID: 1CGZ) and EGCG (PDB ID: 3NG5) were acquired from the Protein Data Bank (PDB) database. The 3D structures of the food-derived compounds CC, RSV and EGCG were subjected to the minimized energy in Chem 3D software using the MM2 forcefield. The missing parts of the PD-L1 dimer were completed by using the WHAT IF server. Subsequently, the docking procedure was carried out utilizing AutoDock Vina [47], in which a grid box of dimensions 20 × 20 × 20 with a grid spacing of 1 Å centered on the binding pocket was established, with other parameters set as default. Validation of the docking method was performed by extracting the inhibitor BMS-200 (PDB ID 5N2F) from the crystal structure and then docking it back into the receptor (PD-L1 dimer). Concurrently, the docking analysis was performed to automatically place CC, RSV and EGCG in the binding pocket of the PD-L1 dimer to obtain initial structures for the MD simulations. Finally, only the conformations with the best binding affinities were selected and the results were visualized using PyMOL software.

### 3.2. Molecular Dynamics Simulation

Following the docking study, all of the complex systems were subjected to MD simulations using the GROMACS 2016.4 package, as described previously by our group [48]. The general AMBER force field (GAFF) [49] was assigned to CC, RSV and EGCG using the Leap module [50]. Amber ff99SB was employed to describe the force field parameters of the PD-L1 dimer [51]. Then, each complex was solvated in a 10 Å cubic box with TIP3P waters. Counterions were added to neutralize the systems, followed by energy minimization (including steepest descent and conjugated gradient) to remove bad contacts. Afterward, the temperature of these systems was increased gradually from 0 to 300 K over 1 ns in the NVT ensemble, and then the pressure was coupled under 1 atm for 1 ns in the NPT ensemble. Finally, 150 ns MD simulations were performed, during which the bond lengths involving hydrogen atoms were constrained using the LINCS algorithm. The short-range nonbonded interactions were computed with a cutoff of 10 Å, and long-range electrostatic interactions were calculated using the Particle Mesh Ewald (PME) algorithm. The temperature was maintained at 300 K using a novel V-rescale thermostat, and the Parrinello–Rahman barostat was used to control the pressure at 1 atm. The simulations were performed thrice with the same parameters to check the stability of the systems and determine the statistical significance of the results. The trajectories were recorded every 1.0 ps for subsequent analysis.

### 3.3. Binding Free Energy Calculation

The stable MD trajectory acquired from each system was chosen to calculate the average binding free energies (Δ*G*) using the MM-PBSA approach [52] implemented in the GROMACS 2016.4 program. Herein, a total of 300 snapshots were sampled from the last 30 ns of MD trajectories at time intervals of 100 ps. Briefly, the MM-PBSA method can be summarized as follows:(1)$${\Delta G=G}_{\mathrm{complex}\;}{-\;(G}_{\mathrm{protein}\;}{+G}_{\mathrm{ligand}})$$
(2)$${G=E}_{\mathrm{MM}}{+G}_{\mathrm{sol}\;}-\;T\Delta S$$
(3)$$E_{\mathrm{MM}}{=E}_{\mathrm{vdw}}{+E}_{\mathrm{ele}}$$
(4)$$G_{\mathrm{sol}}{=E}_{\mathrm{PB}}{+E}_{\mathrm{SA}}$$
(5)$$E_{\mathrm{SA}}=\gamma \cdot \mathrm{SASA}$$
(6)$${\Delta G=\Delta E}_{\mathrm{vdw}}{+\Delta E}_{\mathrm{ele}}{+\Delta E}_{\mathrm{PB}}{+\Delta E}_{\mathrm{SA}}$$

Here, *G_complex_*, *G*_protein_ and *G_ligand_* are the free energies of the complex, protein and ligand, respectively (Equation (1)). The free energy (*G*) in Equation (1) is evaluated as the sum of the gas-phase binding energy (*E_MM_*), the solvation free energy (*G_sol_*) and the entropic contribution (*T*Δ*S*) (Equation (2)). *E_MM_* is further divided into a van der Waals term (*E_vdw_*) and an electrostatic term (*E_ele_*) (Equation (3)). The solvation free energy (*G_sol_*) is further divided into a polar (*E_PB_*) and a nonpolar (*E_SA_*) component (Equation (4)), in which the polar solvation term was calculated using the Poisson–Boltzmann model and the nonpolar term was computed based on the solvent-accessible surface area (SASA) with γ set to the default value (Equation (5)).

However, calculating the entropy (*T*Δ*S*) is computationally expensive, and the inclusion of the entropic contribution in Δ*G* does not always assure the accuracy of binding free energy calculations [53,54]. Therefore, only the relative binding free energy without the entropic effect was evaluated in the present work to determine the binding affinity between the PD-L1 dimer and the food-derived compounds (Equation (6)). To identify the key residues responsible for binding, free energy decomposition to individual residues was performed based on the same snapshots used in the above calculations.

### 3.4. Simulation Analysis

The auxiliary tools provided with GROMACS were employed for trajectory analysis. The DSSP program was used to analyze the secondary structure of the PD-L1 dimer in each system. The mindist program and Protein–Ligand Interaction Profiler (PLIP) were applied to compute the contact numbers and interactions of the CC, RSV and EGCG systems, respectively. The occupancies of intermolecular H bonds in the complex systems were analyzed using Visual Molecular Dynamics (VMD) 1.9.3. software [55] with a common standard; i.e., an acceptor–hydrogen-donor angle >135° and an acceptor–hydrogen atom distance of <3.5 Å.

To characterize the major motion of the PD-L1 dimer and the correlative motion between the atoms derived from the MD trajectories in complex systems, an effective technique of principal component analysis (PCA) was employed [56,57,58]. In the present study, the eigenvectors (namely, the principal components) and the corresponding eigenvalues were produced by addressing the covariance matrix of Cα atoms. The eigenvectors represent the directions of atomic motions, while the eigenvalues describe their corresponding magnitudes. However, a standard PCA method cannot distinguish internal motion from trivial overall motion very well. Thus, PCA was carried out utilizing the backbone dihedral angles of the PD-L1 dimer (dihedral PCA). Then, the first two principal components generated (dihedral PC1 and PC2) were used as reaction coordinates to build an FEL, and the free energy was calculated according to the equation *G* = −*k*_B_*T* × ln*P*, in which *k*_B_ is Boltzmann’s constant, *T* is the temperature of the simulation systems and *P* is the relative probability of the conformational distribution [59,60,61,62].

## 4. Conclusions

In this work, a wide array of computational approaches was employed to present a comprehensive molecular-level picture of the inhibitory mechanism of food-derived polyphenols (CC, RSV and EGCG). Conformational dynamic property analysis showed that the PD-L1 dimer binding to these food-derived compounds generally remained stable throughout the simulation. Analysis of the binding free energy revealed that the ability of these compounds to stabilize the PD-L1 dimer follows the trend of CC > RSV > EGCG; thus, CC is the most effective inhibitor of the PD-1/PD-L1 pathway. Notably, five key residues make crucial contributions to ligand binding, Ile54, Tyr56, Met115, Ala121 and Tyr123, as identified by per-residue energy decomposition. Based on the analysis of binding modes and interactions, three key components were identified: the C, F and G sheets of the PD-L1 dimer. Specifically, the nonpolar interactions between the aromatic rings of such compounds and the key residues of the PD-L1 dimer play a dominant role in enhancing their stability and affinity. This offers an opportunity to identify available food-derived compounds or design new small molecules with similar structural groups that can provide improved binding affinity with the PD-L1 dimer and, thus, effectively inhibit PD-1/PD-L1 interactions. The FEL and DSSP results further imply that these compounds can interact stably with the binding regions of the PD-L1 dimer. Overall, such structural features help to understand the druggable hotspots at the dimer interface and yield insights for developing food-derived molecules that target PD-L1 dimerization, thereby providing a potential cancer immunotherapeutic strategy. On the other hand, it is necessary to remark that, though the binding affinities and the binding pockets of these systems were achieved by MD simulations in this work, further experimental verification is also demanded to provide more comprehensive understandings on the inhibitory mechanism, which will be conducted in the near future.

## Figures and Tables

**Figure 1 ijms-22-10924-f001:**
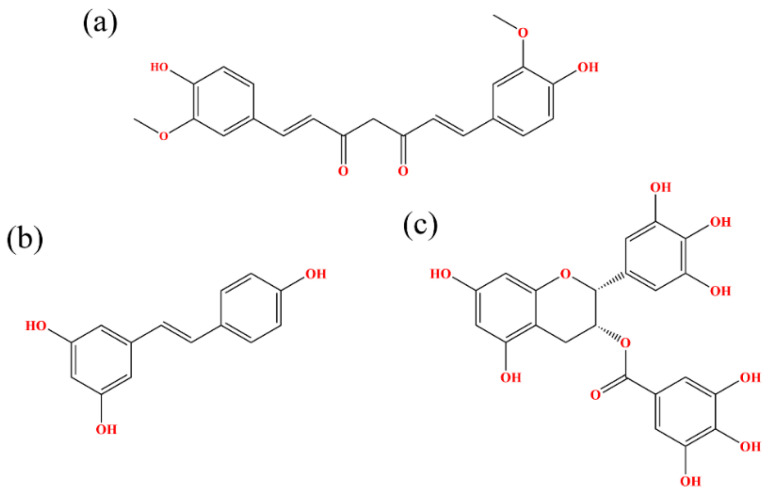
Chemical structural formulas of (**a**) CC, (**b**) RSV and (**c**) EGCG.

**Figure 2 ijms-22-10924-f002:**
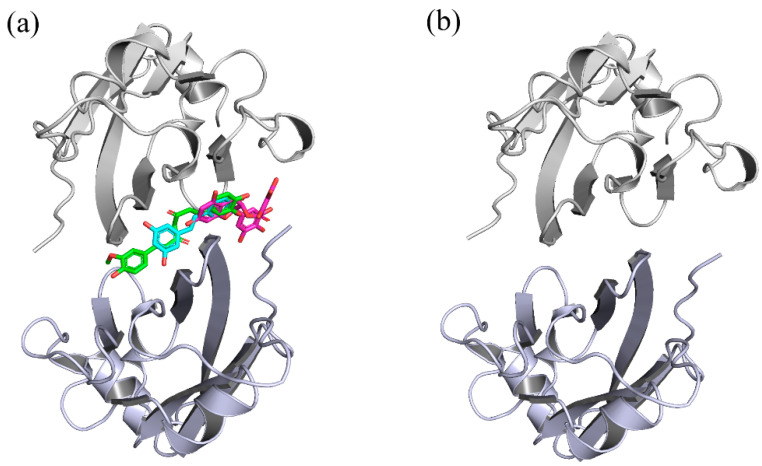
Initial structures of the systems used in MD simulations. (**a**) CC, RSV and EGCG systems. (**b**) dimer system. The CC, RSV and EGCG compounds are shown in green, cyan and magenta, respectively.

**Figure 3 ijms-22-10924-f003:**
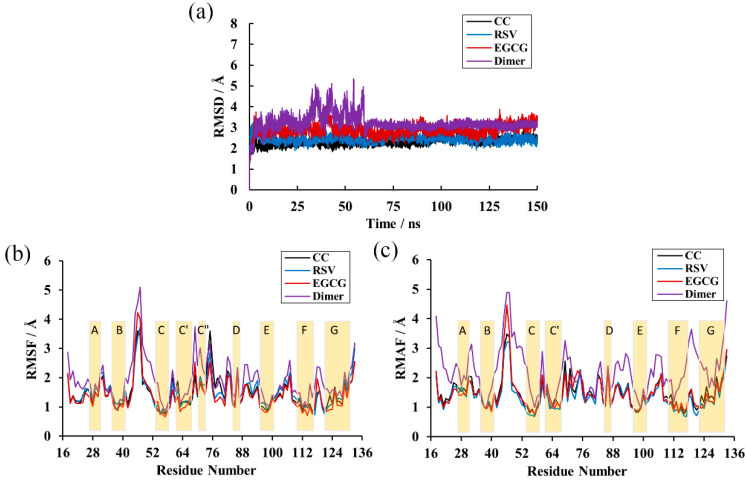
RMSD and Root-mean-square fluctuation (RMSF) results of the MD simulations. (**a**) RMSDs of residues in the CC, RSV, EGCG and dimer systems. (**b**) RMSF fluctuations of residues on _A_PD-L1. (**c**) RMSF fluctuations of residues on _B_PD-L1.

**Figure 4 ijms-22-10924-f004:**
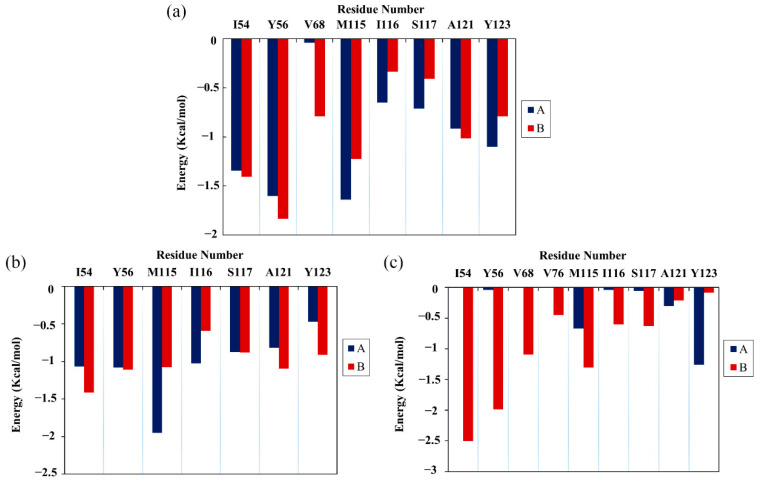
Residue energy decomposition of key residues belonging to the (**a**) CC, (**b**) RSV and (**c**) EGCG systems (kcal/mol).

**Figure 5 ijms-22-10924-f005:**
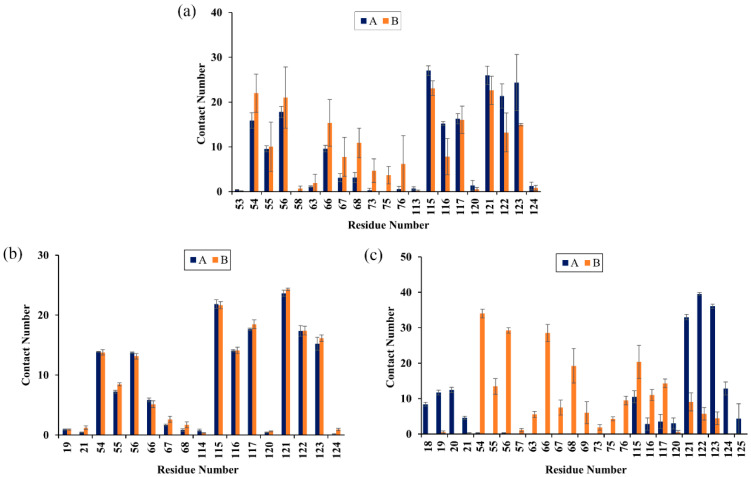
Contact numbers between the food-derived compounds and the PD-L1 dimer in the (**a**) CC, (**b**) RSV and (**c**) EGCG systems. Error bars represent the standard deviations of triplicate calculations.

**Figure 6 ijms-22-10924-f006:**
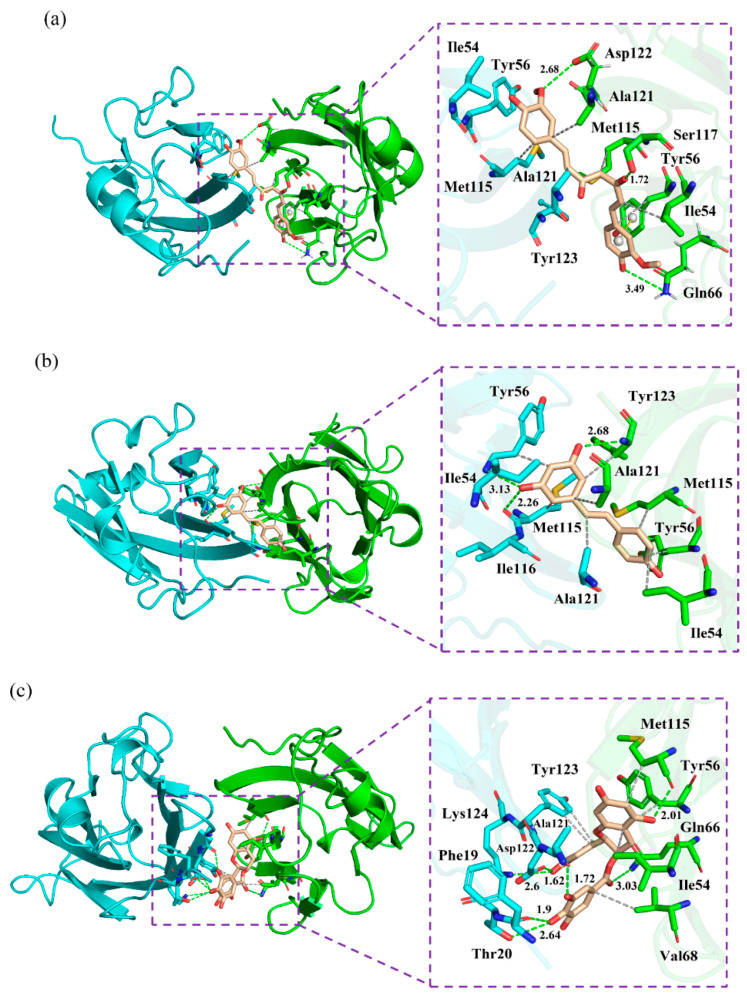
Binding modes of the (**a**) CC, (**b**) RSV and (**c**) EGCG systems. The key residues on _A_PD-L1 and _B_PD-L1 are shown as cyan and green sticks, respectively, while the ligands are shown as beige sticks. Hydrophobic interactions, H bonds and Π-stacking are shown as gray, green and yellow dashes, respectively.

**Figure 7 ijms-22-10924-f007:**
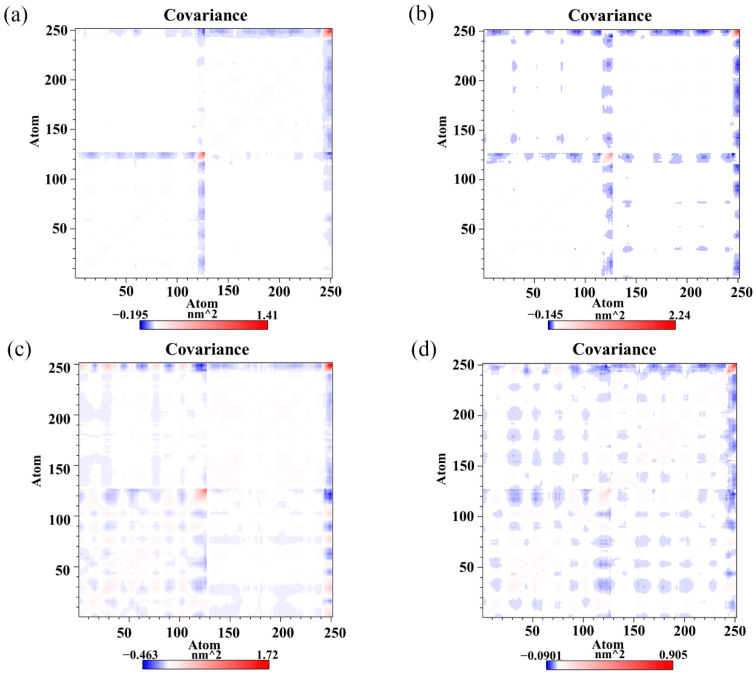
Cross-correlation matrixes of fluctuations in the *x*-, *y*- and *z*-coordinates for Cα atoms belonging to the PD-L1 dimer in the (**a**) CC, (**b**) RSV, (**c**) EGCG and (**d**) dimer systems.

**Figure 8 ijms-22-10924-f008:**
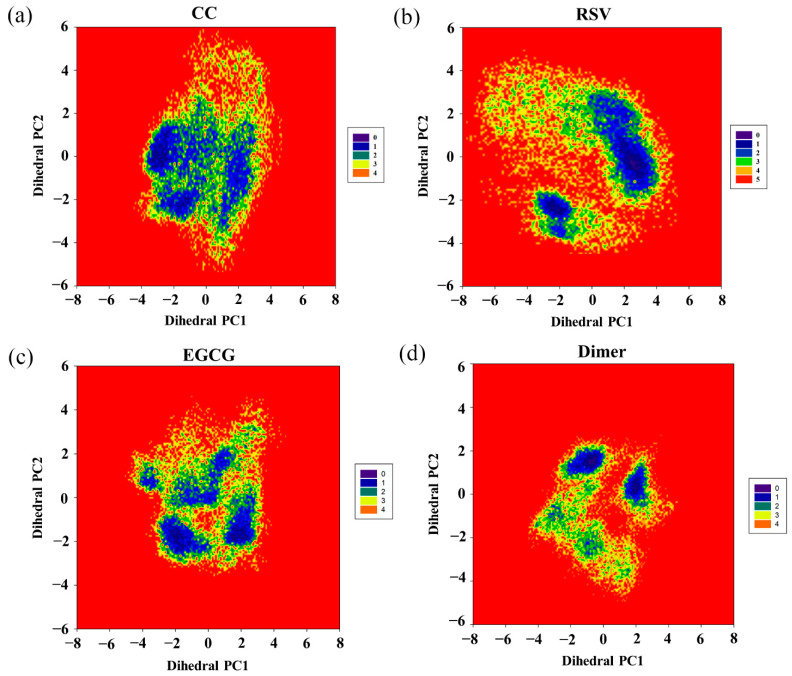
Free energy landscapes (KT) of the (**a**) CC, (**b**) RSV, (**c**) EGCG and (**d**) dimer systems. PC1 and PC2 represent principal components 1 and 2, respectively.

**Figure 9 ijms-22-10924-f009:**
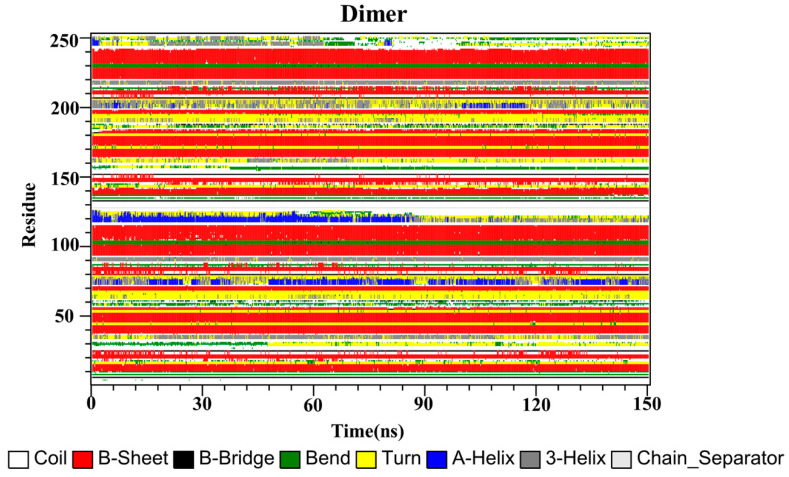
Secondary structures of the PD-L1 dimer in the absence of food-derived compounds.

**Figure 10 ijms-22-10924-f010:**
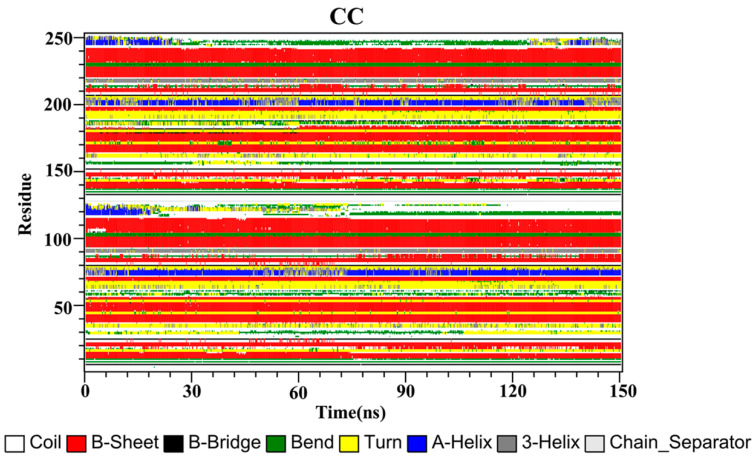
Secondary structures of the PD-L1 dimer in complex with compound CC.

**Figure 11 ijms-22-10924-f011:**
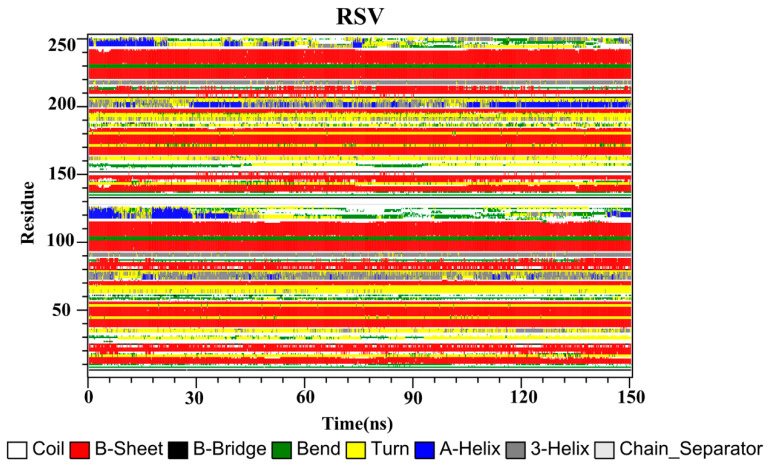
Secondary structures of the PD-L1 dimer in complex with compound RSV.

**Figure 12 ijms-22-10924-f012:**
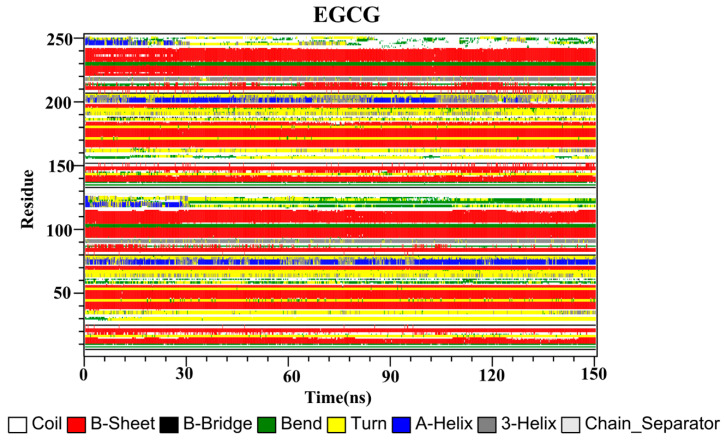
Secondary structures of the PD-L1 dimer in complex with compound EGCG.

**Table 1 ijms-22-10924-t001:** Binding free energies in the CC, RSV, EGCG and dimer systems (kcal/mol).

Contribution	CC	RSV	EGCG	Dimer
Δ*E*_vdw_ ^a^	−57.28 ± 3.27	−42.96 ± 0.40	−42.88 ± 1.70	−44.59 ± 9.85
Δ*E*_ele_ ^b^	−3.83 ± 0.76	−5.16 ± 0.20	−23.14 ± 3.14	−124.35 ± 23.36
Δ*E*_PB_ ^c^	32.33 ± 3.63	23.02 ± 0.22	50.43 ± 3.76	211.28 ± 17.07
Δ*E*_SA_ ^d^	−4.95 ± 0.14	−3.39 ± 0.03	−4.73 ± 0.16	−6.23 ± 0.37
Δ*E*_polar,total_ ^e^	28.50 ± 3.18	17.86 ± 0.42	27.29 ± 1.42	86.94 ± 10.00
Δ*E*_nonpolar,total_ ^f^	−62.23 ± 3.35	−46.35 ± 0.43	−47.61 ± 1.75	−50.82 ± 10.06
Δ*G* ^g^	−33.72 ± 0.23	−28.49 ± 0.40	−20.31 ± 0.35	36.11 ± 0.89

^a^ Van der Waals interaction energy. ^b^ Electrostatic energy. ^c^ Polar solvent effect energy. ^d^ Nonpolar solvent effect energy. ^e^ Polar binding free energy. ^f^ Nonpolar binding free energy. ^g^ Binding free energy. The energies are the average values of the 300 conformations extracted from 120 to 150 ns.

**Table 2 ijms-22-10924-t002:** Contact numbers in the binding domains of the CC, RSV and EGCG systems.

Inhibitor	*N*-Terminal	C Sheet	C’ Sheet	F Sheet	G Sheet	Total Sheet	Total
CC	3	96	50	105	124	376	379
RSV	1	70	18	108	115	311	312
EGCG	32	77	55	63	141	335	368

**Table 3 ijms-22-10924-t003:** Hydrogen bond occupancies of the CC, RSV and EGCG system.

Donor	Donor H	Acceptor	Occupancy (%)
_B_Ser117@OG	HG	CC@O1	76.74
CC@O2	H7	_B_Gln66@OE1	87.04
RSV@O3	HO3	_A_Met115@O	76.74
EGCG@O50	H50	_A_Ala121@O	80.40
EGCG@O10	H10	_A_Asp122@N	59.80
EGCG@O47	H47	_A_Phe19@O	59.47
EGCG@O10	H10	_A_Asp122@OD1	39.53
EGCG@O03	H03	_B_Met115@O	43.52

## Data Availability

The data presented in this study are available within the article, figures, tables and Appendix A.

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
