# Peer review of "Molecular Mechanism of Food-Derived Polyphenols on PD-L1 Dimerization: A Molecular Dynamics Simulation Study"

_ijms, 2021, doi:10.3390/ijms222010924_

Round 1

Reviewer 1 Report

The submitted manuscript is a computational study based on the hypothesis developed by Verdura et al. (cited by the authors as reference 28) on the inhibitory mechanism of RSV. According to Verdura et al. model, RSV can operate as a direct inhibitor of glyco-PD-L1-processing enzymes and through a non-mutually exclusive mechanisms of binding to the dimerization surface of PD-L1, targeting the same interface employed by PD-L1 to interact with PD-1. In the submitted work, the authors applied a similar computational protocol to study the binding of RSV and of two ligands, EGCG and CC, to the same pocket. However, no experimental evidences were reported by the authors to support that the EGCG and CC might affect/inhibit the PD-L1 dimer or bind to the same pocket of BMS-202. The lacking of such experimental validation is the major limit of the manuscript that remains too speculative.

In addition, since the prediction of the free energy of binding with the MM-GBSA method depends on several computational parameters and different settings may result in very different binding affinities even for the same system in study, the lacking of information for these systems makes the calculations more critical. Thus, I encourage the authors to include as much experimental support as possible

Reviewer 2 Report

 The authors have submitted a MS to demonstrate the mechanism by which the food-derived polyphenol down-regulate of PD-L1 expression. Their results show that polyphenols inhibit PD-1/PD-L1 interactions targeting PD-L1 dimerization and pave a new avenue to develop novel immunomodulatory molecules

The MS is interesting and clear and have no major comment   

Reviewer 3 Report

I have completed the review of the article: ''Molecular Mechanism of Food-Derived Polyphenols on PD-L1 Dimerization: A Molecular Dynamics Simulation Study.''
It is an interesting study that uses natural derived polyphenolics in the treatment of cancer. Results showed that the use of curcumin, resveratrol, and epigallocatechin gallate have the ability to interrupt the PD-1 / PD-L1 pathway by targeting PD-L1 dimerization, thus these compounds that can be naturally found in food can be useful in treating cancer. The paper is written clearly and concisely, the figures are qualitative and the tables are easy to follow. The methods used (molecular docking, simulations and MM-PBSA calculations) are briefly described so that they can be easily reproduced. I consider that the paper meets all the conditions for publication.

Reviewer 4 Report

Manuscript is well designed and experiment was well conducted.

Results are fairly presented.

Some comments are addressed to:

Lines should have been numerated in order.  

Page 1.

Use impersonal style of writing. Avoid we, our...

Apply this though the whole document.

Page 3.

Before mentioning Figure S1 and Figure S2, authors should describe content of Figures 1 and 2 presented in the main text.

Page 7.

A raw should be inserted between the Table and the paragraph.

Page 9.

A raw should be inserted between the Table 3 and the paragraph.

 Detailed comments are included in the PDF document.

Round 2

Reviewer 1 Report

I have no additional suggestions